# Namibian Healthcare Professionals’ Knowledge, Attitudes and Practices Regarding Environmental Sustainability in Healthcare

**DOI:** 10.3390/ijerph22050751

**Published:** 2025-05-09

**Authors:** Helga Elke Lister, Karien Mostert, Gopika Ramkilawon, Cathrine Oelschig, Olwethu Ntiyane, Erika Richardt, Deonelia Paulo Da Silva Rocha, Savannah Sheerin, Tshepang Phaahla, Daniel Ashipala, Louise Pretorius, Takaedza Munangatire, Filip Maric

**Affiliations:** 1Department of Occupational Therapy, Faculty of Health Sciences, University of Pretoria, Pretoria 0028, South Africa; helga.lister@up.ac.za (H.E.L.);; 2Department of Physiotherapy, Faculty of Health Sciences, University of Pretoria, Pretoria 0028, South Africa; 3Department of Statistics, Faculty of Natural and Agricultural Sciences, University of Pretoria, Pretoria 0028, South Africa; 4Department of Nursing Sciences, School of Nursing and Public Health, University of Namibia; Windhoek 13301, Namibia; 5Department of Health and Care Sciences, Faculty of Health Sciences, UiT The Arctic University of Norway, 9020 Tromsø, Norway

**Keywords:** knowledge, attitudes and practices, environmental sustainability, healthcare professionals, Namibia, Africa, sustainable healthcare, planetary health, healthcare leadership, low- and middle-income countries, climate change

## Abstract

Among the many actions required to avert further intensification of today’s social, ecological and health crises is also the improvement of healthcare’s environmental sustainability, including in countries particularly vulnerable to such crises. The present study aimed to identify Namibian healthcare professionals’ knowledge, attitudes and practices, along with barriers and educational needs, as a foundation for context-relevant interventions. The study used a non-experimental, descriptive quantitative research design with an existing validated cross-sectional questionnaire as its data collection tool. Both purposive and snowball sampling were used to select healthcare professionals (*n* = 71) to participate in the quantitative online questionnaire. R (version 4.2.1) software was used to analyse the data from the completed questionnaires. The results showed that the Namibian healthcare professionals participating in this study have basic knowledge of and positive attitudes toward environmental sustainability in healthcare. However, various barriers to implementing strategies towards environmental sustainability exist that currently prevent the implementation of relevant practices. These should be overcome by the Namibian health system by providing the necessary frameworks, policies, measures and resources to drive improvements in environmental sustainability. Additionally, future and current healthcare professionals must receive training across all professional education levels to enable implementation in practice and effective advocacy and planetary health promotion.

## 1. Introduction

Today’s world is marked by rapidly intensifying social, ecological and health crises. Human-driven climate change, nature and biodiversity loss and pollution and waste are deeply intertwined with and exacerbate human crises, including deteriorating health, conflict for territory and resources and displacement [1]. While this makes virtually any country around the world vulnerable to significant disruption, especially so in today’s political climate, regions like sub-Saharan Africa are already experiencing significant shifts and higher vulnerability, not least due to the complex intersections between socioeconomic, political, infrastructural and environmental factors [2,3,4].

It is now widely known that healthcare systems and services, in contradiction to their aspirations to do no harm, contribute to ecological degradation via greenhouse gas emissions, waste production, pollution and other pathways [5]. Internationally, this paradoxical situation is increasingly met through efforts to improve its environmental sustainability via both top–down and bottom–up initiatives [6]. Because progress in healthcare environmental sustainability also requires participation from stakeholders from across and beyond the health sector, one critical ingredient for effective implementation of relevant strategies is to understand the current knowledge, attitudes and practices of all stakeholders. Studies to understand the relevant knowledge, attitudes and practices of healthcare professionals are increasingly being conducted around the world, although for the most part, in Global North contexts [7,8,9].

There are, however, challenges when considering translocating this kind of work to Global South contexts. Firstly, not all countries bear equal responsibility for ecological degradation. There is, rather, often an inverse relationship between climate change contributions and climate effects and vulnerability [10,11].

Taking Namibia as an example, the country in focus in the present study, is a net carbon sink at present and is projected to remain so in 2030, with its share of global aggregated emissions weighing a mere 0.00026% [12]. At the same time, Namibia is highly vulnerable to climate change and already feeling its impacts through persistent droughts, a higher frequency of floods and epidemics, intertwining with rapid population growth, climate migration and urbanisation, poverty, inequalities and socioeconomic marginalisation, all with likely effects on further environmental degradation and effects on the health of local communities [13,14]. Regardless of its comparatively small contribution to climate change, Namibia’s government is concerned about ongoing ecological degradation and is strategising national mitigation and adaptation across all sectors [12].

Secondly, although there is somewhat limited data on the environmental footprint of healthcare in low-income countries, healthcare environmental footprints are typically small, with possible relationships to overall insufficient health provision, low per-capita healthcare expenditure and overall low population health [5,15]. The extent to which healthcare systems and professionals in countries like Namibia have cause or responsibility to attend to questions of healthcare environmental sustainability might, therefore, be comparatively low.

Yet, findings from a study on healthcare professionals’ knowledge, attitude and practices on environmental sustainability conducted in immediately neighbouring South Africa indicated that healthcare professionals had positive attitudes and a high level of interest in being educated on environmental sustainability, its implementation within healthcare and taking on increased responsibility in this sector [8]. This, at least, gives the first indication of a general interest and willingness to contribute to the subject field among Southern African healthcare professionals. Given that health professionals have additionally been called to be leaders of change in advocating for environmentally sustainable practices, their buy-in (and by implication, the buy-in of their patients) would make significant contributions to local environmental sustainability and, therefore, to the health of the population [16,17,18,19,20,21].

To determine how Namibian healthcare professionals can advance environmental sustainability in healthcare and how government and local organisations can facilitate this, it is necessary to gain insight into healthcare professionals’ current knowledge, attitudes and practices of environmental sustainability in healthcare.

## 2. Materials and Methods

### 2.1. Research Design

The aim of the study was to determine the Namibian healthcare professionals’ current knowledge, attitudes and practices of healthcare-related environmental sustainability, as well as perceived barriers to its improvement in the Namibian healthcare sector. The study used a non-experimental, descriptive quantitative research design with an existing validated cross-sectional questionnaire as its data collection tool [22].

### 2.2. Sampling and Sampling Technique

The participating healthcare professionals were occupational therapists, physiotherapists, nurses, registered dieticians, audiologists, speech therapists and dual-registered speech therapists and audiologists. All participating healthcare professionals were living in Namibia, registered with the Health Professions Council of Namibia (HPCNA) and had practised in their respective fields for at least six months. At the time of the study, there were 12,262 healthcare professionals in Namibia. Information about the study and the link to the questionnaire were disseminated through various professional organisations and academic channels. Additionally, the researchers identified and distributed a link for the questionnaire to numerous healthcare professionals via email and social media (mostly WhatsApp and Facebook) using purposive sampling. Snowball sampling followed after that, as the selected healthcare professionals distributed the questionnaire to further relevant healthcare professionals who met the inclusion criteria. Despite these extensive recruitment strategies, only 71 (0.58%) healthcare professionals completed the questionnaire.

### 2.3. Study Tool and Data Collection

The questionnaire consisted of five sections covering (1) knowledge, (2) attitudes, (3) practices, (4) barriers and (5) education. The questions ranged from true and false questions to closed- and open-ended questions. The questionnaire has content validity in 9 of 11 Southern African countries (including Namibia), with a scale-content validity Index/Average (S-CVI/Ave) of 0.922.

The questionnaire was uploaded onto Qualtrics, (version XM/os2) an online software platform, from which a link was sent out to participants to complete the questionnaire electronically. Qualtrics uses Transport Layer Security encryption that keeps all transmitted data confidential. Only those with relevant access can view the results of completed questionnaires. Ethical clearance was received from the Ethics Committee, Faculty of Health Sciences, University of Pretoria (reference number 648/2022), and the Namibian Ministry of Health Research Committee. Informed consent was obtained from all participants. Participants provided informed consent before completing the questionnaire, were not remunerated and could withdraw from the study at any stage. The questionnaire was made available to complete over a ten-week period.

## 3. Results

### 3.1. Demographic Characteristics

As seen in Table 1, over half of the respondents were nurses (62.86%), while approximately one-quarter comprised physiotherapists (12.86%) and occupational therapists (11.43%). Audiologists, registered dieticians, speech therapists and audiologists comprised the smallest groups of healthcare professionals represented (between 1% and 5%). The majority of the respondents were working in private (38.57%) and public sectors (35.72%), while one-quarter were employed in academia (24.29%), and just over 1% were working in non-governmental sectors. It should be noted that the respondents were able to have multiple work roles, so the percentages are represented per the number of respondents and do not sum to 100. Three-quarters of respondents worked in a clinical role (70.27%), while one-third worked in education (36.49%) and another third in management and research (35.13%). The vast majority of participants were female (78.57%) compared to a smaller number of male participants (21.43%). The mean age range for participants in the study was 40.23 years (ranging from 37.62 to 42.84).

As seen in Table 2, the most common home language spoken among participants was Oshiwambo (34.29%). This was followed by English (18.57%), Afrikaans (11.43%) and 35.71% selected other languages. Thirty-seven (52.86%) participants were able to speak excellent English, while thirty-three (47.14%) spoke good English.

### 3.2. Knowledge (n = 68)

Sixty-eight of seventy-one participants responded to this section. The participants were asked to respond to various questions about knowledge relating to climate change and health. As seen in Table 3, the questionnaire items presented to the participants ranged from “*climate change has an impact on human health*” to “*empowering healthcare professionals to practise environmentally sustainable healthcare has far-reaching benefits*”. Participants were given four options to choose from: *true, false, I don’t know* and *not applicable*. The responses to the questionnaire items are such that true responses are correct, and false responses are incorrect [15]. The “*not applicable*” option was to be selected if the participant did not agree that climate change is happening (refer to Table 3 for the responses). The lowest percentage of participants who answered correctly on a questionnaire item was 77.94%, while for the other items, between 1.47% and 14.71% of participants responded incorrectly, and between 1.47% to 16.18% of participants reported that they did not know whether the item was correct or incorrect. No participants selected “*not applicable*”.

When asked to select answers from various options relating to the question “*Climate change has a direct negative impact on human health, through which of the following?*”, more than 50% of participants selected heat-related illnesses, illnesses related to air pollution, the loss of livelihood, self-care and meaningful leisure pursuits, malnutrition and waterborne illnesses. (Refer to Table 4.) Only one participant did not agree that climate change is happening.

### 3.3. Attitudes (n = 66)

Sixty-six of seventy-one participants responded to this section. The participants were asked to respond to the questions based on their attitudes towards environmental sustainability in healthcare using a five-point Likert scale ranging from strongly disagree to strongly agree. If the percentage difference between strongly agree and somewhat agree was below 40%, the responses were grouped together, similarly for strongly disagree and somewhat disagree.

The percentages for disagreeing with all the items in the second section of the questionnaire ranged between 7.58% and 10.61%. Over half of the participants (65.15%) responded that they agree with the item “*I am concerned about the contribution of the healthcare industry to resource depletion*”, with 25.76% being neutral. Similarly, more than half of the participants (72.97%) agreed they were concerned about the impact of climate change on human health and wellbeing, with 6.76% being neutral. Most participants (86.36%) agreed that environmental sustainability practices should be incorporated into healthcare services (and of those, 65.15% strongly agreed, while 21.21% somewhat agreed), with 6.06% being neutral. Additionally, most of the participants (more than 80%) felt that healthcare professionals have a professional obligation to contribute to environmental sustainability and that their actions in their profession can contribute positively to environmental sustainability, with 6.06% and 7.58% of the latter items being neutral.

### 3.4. Practices (n = 65)

Sixty-five of seventy-one participants responded to this section. The participants had the opportunity to answer questions regarding existing environmentally sustainable practices within their place of work or the strategies that they would support. For the first question, the participants were asked whether their place of work has a policy that addresses or incorporates environmental sustainability. Only 29.23% of participants responded yes, 41.54% responded no, and 29.23% responded that they did not know of any policies.

For the following question, the participants could choose more than one strategy that has been implemented at participants’ places of work to improve environmental sustainability, regardless of whether their place of work had a policy or not. More than 40% of participants selected the following strategies: *implementation of clear procedures for handling and disposing of medical waste* (58.11%), *recycling programmes* (48.65%), *reducing the use of paper and printing* (47.3%), *recycling procedures and recycle bins* (43.24%) and *reusing items where possible* (41.89%). The least number of participants selected the strategy *using dry-composting toilets instead of water-flush toilets* (2.7%).

The participants were then asked if they consider environmental sustainability when prescribing, using or choosing equipment and materials as a healthcare professional. More than half of the participants (67.69%) answered that they sometimes considered this while working, while 21.53% selected always/most of the time, and 10.77% selected never. 

When asked whether the participants were interested in implementing strategies in their place of work that can contribute to environmental sustainability, almost all the participants (96.92%) said “yes”. They were then provided with a list of potential environmentally sustainable measures at their place of work that they would support. More than half of the participants selected that they would support the *training of healthcare professionals on the importance of environmentally sustainable practices* (67.61%), *reduced use of paper and printing* (63.38%), *implementation of a recycling programme* (61.97%), *advocating for the implementation of environmentally sustainable practices* (60.56%) and *spending money on sustainable items/equipment* (56.34%). In addition to this, half of the participants stated that they would support *choosing suppliers that adhere to standards that promote a cleaner environment* (52.11%), *green energy supplies* (46.48%) and the *implementation of clear procedures for handling and disposing of medical waste* (52.11%) as environmentally sustainable measures. 

Most (96.88%) of the participants acknowledged that healthcare professionals have a role *in taking action towards environmental sustainability*. On being asked in which ways healthcare professionals could take action, the most frequently selected choices were *leading by example in practising environmental sustainability* (70.42%), *raising awareness of climate change as a universal health matter* (66.2%) and *advocating for mitigation strategies in the health sector* (63.38%). Furthermore, a suggestion was made by one participant to *incinerate medical waste to aid in reducing air pollution*.

### 3.5. Barriers (n = 61)

Sixty-one of the seventy-one participants responded to this section. The participants could choose more than one barrier from a predetermined list and specify other barriers that were not included in the list. The main barriers that *prevent healthcare professionals from implementing environmentally sustainable strategies* that the participants selected included a lack of knowledge (52.11%), time (45.01%), skills (40.85%) and support from colleagues (36.62%). Other barriers included that it costs too much (18.31%) and that it requires too much effort (14.08%). Seven respondents (9.86%) indicated that it was not a priority. The additional information provided indicated that changing the traditional mindsets of community members is challenging and prevents the implementation of environmentally sustainable practices.

The main barriers that *prevent healthcare systems from implementing environmentally sustainable strategies*, as noted by the participants, included limited resources (61.97%), an increased demand for healthcare services (60.56%), a lack of policy or guidelines (59.15%) and shortage of healthcare staff (59.15%). The following selected responses as challenges preventing healthcare systems from implementing these practices were all between 25% and 50%: *the global economic crises* (39.44%), *lack of infrastructure* (39.44%), *increased healthcare costs* (36.62%), *higher patient expectations* (26.76%) and *increased employee turnover* (26.76%). Responses under “*other (please specify)*” included ignorance regarding climate change, the mismanagement of funds and a lack of priorities as further barriers. 

The participants had to select what *would help implement environmentally sustainable strategies*. The main strategies selected were education on climate change and its impact on health (67.61%); specific policies in their place of work (64.79%); training to be able to communicate effectively about climate change and health to their colleagues and patients (60.56%); collaboration amongst others (57.75%); direct guidance by environmental sustainability experts on how to make the workplace environmentally sustainable (57.75%); financial support (54.93%) and having access to resources (54.93%). Other selected options were that healthcare professionals *already work in a demanding profession and do not have enough time, energy and compassion to consider environmentally sustainable practices*.

### 3.6. Education (n = 58)

Fifty-eight participants completed this section. First, the participants were asked to complete a Likert scale rating on the statement that “*educational input should be given within your profession regarding environmental sustainability in healthcare*”. More than half of the participants (74.14%) agreed with the statement that “*educational input should be given within your profession regarding environmental sustainability in healthcare*”, while 13.79% of the participants responded that they did not agree with the statement, and 12.07% remained neutral.

The second questionnaire item was about their preferred method of being educated with regard to the topic. The participants selected continuous professional development activities (59.15%), followed by online interactive events (e.g., webinars) (49.3%), the provision of online learning materials by instructors that can be accessed at any time (46.68%) and in-person workshops (42.25%). Other preferred methods include online videos (33.38%), social media platforms (36.62%), pamphlets or books (30.99%), community-based project learning (26.76%) and emails, while websites (19.72%) and theatrical performances (15.49%) were the least selected. 

### 3.7. Other (n = 8)

Eight participants completed this section. The participants could add further comments at the end of the questionnaire. Of the eight participants who chose to do so, most reiterated their responses from preceding sections, including the need for the improvement and better integration of environmental sustainability practices in all settings and the education of healthcare professionals.

One respondent brought to light the use of a top–down approach in addressing policies and the implementation of environmental sustainability practices. The respondent stated that a bottom–up approach is “unfair and unrealistic” for healthcare professionals, due to the expense of time and resources from healthcare professionals.

## 4. Discussion

The findings from our study of healthcare professionals on their knowledge, attitudes and practices regarding environmental sustainability in Namibia resonate closely with a recent study conducted with healthcare professionals in South Africa and the other literature in the field [7,8,9].

Our study confirms that healthcare professionals in Namibia have basic knowledge regarding the impact of climate change on human health and the environment. However, the extent of this knowledge and their understanding is unknown, since their responses must be interpreted in light of the questionnaire providing participants with closed-ended questionnaire items responses as opposed to open ended. Since the participants further indicated their major barrier to implementing environmentally sustainable practices was a lack of knowledge, this study, therefore, indicates that education on environmental sustainability in relation to healthcare appears necessary.

Also aligning with the literature [23,24], our study participants’ attitudes reflected concerns regarding the negative impact of climate change on human health and wellbeing, a positive attitude toward implementing environmentally sustainable practices in healthcare and agreement over a professional responsibility to do so. Yet, a quarter of participants also chose to remain neutral towards the statement “*I am concerned about the contribution of the healthcare industry to resource depletion*”, suggesting a possible lack of knowledge on this aspect. Equally correlating with the literature, our study participants also perceived a responsibility to advocate for and implement environmental sustainability practices by educating people and leading by example [16,17,18,19,20,21]. They also felt that their actions could contribute positively to the environment and environmental sustainability. This shows a willingness to take ownership and responsibility among participating healthcare professionals concerning environmental sustainability in healthcare.

Most participants in our study indicated that they only consider environmental sustainability occasionally/sometimes in practice yet identified with a strong desire and need to do so more. Interestingly, only half of the participants reported that they would support sustainable suppliers and use their sustainably sourced supplies, and only a few selected improving incinerators and promoting waste-to-energy incineration practices, energy-efficient systems and green energy supply as concrete strategies already in practice. This finding may reflect that the participants buy supplies for their departments; however, in general, they are not directly involved in the other elements, such as incineration. Additionally, it may indicate a lack of understanding of these suggested practices.

Our study highlights the barriers that healthcare professionals in developing countries in Africa, such as Namibia, face with regard to the implementation of environmental sustainability in healthcare, including potential strategies to overcome them. At the level of health professionals, a lack of knowledge, skills and support from colleagues could, as highlighted by our study participants and relevant literature alike, potentially all be addressed through the provision of relevant education with regard to concrete practices, climate and health communication strategies and more. At higher levels, our study participants highlighted the lack of targeted policies and guidelines as a major barrier. The current Namibian Nationally Determined Contribution document makes provision for environmental sustainability policies on a multisectoral level, with only one measure for the healthcare sector, to specifically improve health security [12]. In line with the other literature on such regional challenges [25], the development of targeted policies and guidelines was consequently noted as a critical strategy to overcome this barrier, with the additional note that their development needs to be driven from the top down and supported with enabling time, staff, infrastructure and financial resources [26].

Education is an integral part of implementing environmental sustainability practices in healthcare. Our study shows that over half of the respondents are eager to learn about the impacts their interprofessional practices have on the environment. This is in line with literature that highlights the importance of interprofessional education in this field [27]. Our results also support claims that curricula and educational courses taught to health professionals should support environmentally sustainable change [26]. Most of our participants favoured learning through continuous professional development activities, followed by in-person workshops and online interactive events.

Future research should investigate the knowledge, attitudes and practices of healthcare professionals in other countries in Southern Africa, which may have a different landscape to Namibia, specifically from a population and climate impact perspective. Such research could provide a more comprehensive impression of the region and a stronger foundation to develop innovative responses to the global call for environmental sustainability in healthcare. For Namibia itself, we recommend that research should now focus on implementation, preferably driven by relevant policies and involvement from different leadership levels, with close consideration of national and local cultures and contexts.

### Study Limitations

The questionnaire was released without forced completion per question, allowing the respondents to move on without completing a question. This resulted in only 58 participants completing over 90% of the questionnaire. This study did not seek to include a representative sample of the entire population, since the authors were unable to obtain the information of all the healthcare professionals (or a random sample) to ensure representation. Therefore, the results cannot be generalised to all healthcare professionals in Namibia. This potential for selection bias arose from the use of purposive and snowball sampling methods. Even so, the questionnaire was sent to all Namibian health professional organisations, requesting dissemination to their members. Additionally, the questionnaire was validated for audiology, dietetics and human nutrition, nursing, occupational therapy, physiotherapy and speech-language pathology and not for other health professionals, for example, doctors [22]. It would have been valuable to also include these professions in the study, since they may have different views from those included here.

## 5. Conclusions

This study aimed to identify Namibian healthcare professionals’ knowledge, attitudes and practices, along with barriers and educational needs, regarding environmental sustainability in healthcare. This study was conducted to establish foundations for context-relevant interventions to further environmental sustainability in Namibia’s healthcare system and services. The need for clearly defined and specific policies and guidelines from governing bodies aimed at Namibia’s healthcare system was identified. It is suggested that healthcare professionals work collaboratively with policymakers to enhance environmental sustainability practices. It is also advised that regulatory bodies should be implemented and held responsible for ensuring these policies are followed. Finally, we recommend that the education of healthcare professionals includes aspects of environmental sustainability and planetary health, both in undergraduate and postgraduate curricula, as well as continuous professional development. Our findings emphasise the need to integrate policies with practices and enhance training and education for healthcare professionals to improve environmental sustainability in healthcare in Southern Africa.

## Figures and Tables

**Table 1 ijerph-22-00751-t001:** Demographic information.

Profession	Population (%) (*n* = 12,296)	Participant Count (%) (*n* = 71)
Audiology	6 (0.05%)	2 (2.86%)
Dietetics	34 (0.28%)	3 (4.29%)
Nursing	12,012 (97.69%)	44 (62.86%)
Occupational therapy	84 (0.70%)	8 (11.43%)
Physiotherapy	139 (1.13%)	9 (12.86%)
Speech therapy	11 (0.09%)	3 (4.29%)
Dual speech therapy and audiology	10 (0.08%)	1 (1.43%)
Work Sector	Participant Count (%) (*n* = 71)	
Academia	17 (24.29%)	
Non-governmental	1 (1.43%)	
Private	27 (38.57%)	
Public	25 (35.71%)	
Missing value	1	
Work Role	Participant Count (%) (*n* = 71)	
Clinical	52 (73.24%)	
Educational	30 (42.25%)	
Management	16 (22.54%)	
Research	10 (14.08%)	
Other	1 (1.41%)	
Age	Years	
Mean (SD)	40.23 (10.93)	
Range	21–67	

**Table 2 ijerph-22-00751-t002:** Home language and proficiency in English.

Participants’ Language	Participant Count (%) (*n* = 71)
Oshiwambo	24 (32.29%)
English	13 (18.57%)
Afrikaans	8 (11.43%)
Other languages	25 (35.71%)
Missing values	1 (1.41%)
Proficiency in English	Participant Count (%) (*n* = 71)
Excellent	37 (52.86%)
Good	33 (47.14%)
Missing values	1 (1.41%)

**Table 3 ijerph-22-00751-t003:** Knowledge.

	Participant Count (%) (*n* = 68)		
Question	True	False	I Don’t Know
Climate change has a direct negative influence on human health	66 (97.06%)	1 (1.47%)	1 (1.47%)
Climate change has an impact on mental health	62 (91.18%)	3 (4.41%)	3 (4.41%)
If reduction and recycling of medical waste can decrease the impact of climate change and its effects on people’s health	57 (83.82%)	3 (4.41%)	8 (11.76%)
The process of production, transport and use of medical equipment contribute to climate change	53 (77.94%)	4 (5.88%)	11 (16.18%)
Poverty-stricken countries are particularly vulnerable to climate change and environmental degradation	53 (77.94%)	10 (14.71%)	5 (7.35%)
Empowering healthcare professionals to practice environmentally sustainable healthcare has far-reaching benefits	58 (85.29%)	2 (2.94%)	8 (11.76%)

**Table 4 ijerph-22-00751-t004:** Knowledge.

Question	Participant Count (%) (*n* = 68)
Climate change has a direct negative	
impact on human health, through which of the following?	
Select all that apply:	
Heat-related illnesses	49 (66.22%)
Illnesses related to air pollution	61 (82.43%)
Infectious diseases	29 (39.19%)
Loss of livelihood, self-care and meaningful leisure pursuits	43 (58.11%)
Malnutrition	46 (62.16%)
Vector-borne illnesses	33 (44.59%)
Waterborne illnesses	47 (63.51%)
Other	1 (1.35%)
I do not agree that climate change is happening	1 (1.35%)

## Data Availability

The raw data supporting the conclusions of this article will be made available by the authors on request.

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
