# Peer review of "Namibian Healthcare Professionals’ Knowledge, Attitudes and Practices Regarding Environmental Sustainability in Healthcare"

_ijerph, 2025, doi:10.3390/ijerph22050751_

Round 1
Reviewer 1 Report
Comments and Suggestions for Authors
The manuscript highlights the importance of assessing the knowledge, attitudes and practices of health workers about environmentally sustainable practices in healthcare. This is important to bring change to the healthcare system in order to make achieve environmental sustainability.
unfortunately, only 0.58% of all registered healthcare workers that the authors targeted participated in the study. From this small sample the authors did find a basic knowledge and positive attitudes towards environmental sustainability in healthcare. However, it could be that healthcare workers that were already interested in this topic participated. This should be discussed some more in detail.
Reviewer 2 Report
Comments and Suggestions for Authors
I have commented in the attached file

Reviewer 3 Report
Comments and Suggestions for Authors
Dear authors,
I found your paper interesting and wellwritten on a different and important topic. Can not recall I have read any paper covering that topic, so congrats. Enjoyed reading it also for a totally diferent reason, the fact that I have family both in South Africa and Namibia, however no conflict of interest...
I have a couple of comments only. My main concern is that I feel you should motivate why you chose this sampling technique, instead of drawing a random sample from each professional category. You point out this as a study limitation yourselves, not enabling a generalisation of your findings to all health professionals in Namibia. Hence, a potential selection bias.
Then, I have a minor comment or rather question on the top of page 4 and Table 1 on the participant counts on Work role, as the percentages add up to >100%. That is, did several participants have two roles or more? I would suggest you to clarify this.
Furthermore in the same part of Table 1, I am puzzled by "27 (36.49%16 (21.62%)", so something you should clarify or rather correct, I assume.
Then, you state in the text body that another third are working in management and research (35.13%), but when adding up the corresponding percentages in Table 1 I get 18.92% only.
